# Aerobic Exercise-Induced Changes in Cardiorespiratory Fitness in Breast Cancer Patients Receiving Chemotherapy: A Systematic Review and Meta-Analysis

**DOI:** 10.3390/cancers12082240

**Published:** 2020-08-11

**Authors:** Guilherme Maginador, Manoel E. Lixandrão, Henrique I. Bortolozo, Felipe C. Vechin, Luís O. Sarian, Sophie Derchain, Guilherme D. Telles, Eva Zopf, Carlos Ugrinowitsch, Miguel S. Conceição

**Affiliations:** 1Department of Obstetrics and Gynecology, Faculty of Medical Sciences, University of Campinas, Campinas, São Paulo 13083-881, Brazil; guilherme.maginador@gmail.com (G.M.); hbortolozo@gmail.com (H.I.B.); sarian@unicamp.br (L.O.S.); sophie.derchain@gmail.com (S.D.); 2School of Physical Education and Sport, University of São Paulo, São Paulo 05508-030, Brazil; manelix.ef@gmail.com (M.E.L.); felipe.cassaro@yahoo.com.br (F.C.V.); guitelles11@hotmail.com (G.D.T.); ugrinowitsch@gmail.com (C.U.); 3Department of Exercise Oncology, Mary MacKillop Institute for Health Research, Australian Catholic University, Melbourne 3000, Australia; Eva.Zopf@acu.edu.au; 4Faculty of Physical Education, University of Campinas, Campinas 13083-851, Brazil

**Keywords:** aerobic fitness, breast cancer, exercise rehabilitation, VO_2max_

## Abstract

While performing aerobic exercise during chemotherapy has been proven feasible and safe, the efficacy of aerobic training on cardiorespiratory fitness (CRF) in women with breast cancer undergoing chemotherapy has not yet been systematically assessed. Therefore, the objective of this work was to determine (a) the efficacy of aerobic training to improve CRF; (b) the role of aerobic training intensity (moderate or vigorous) on CRF response; (c) the effect of the aerobic training mode (continuous or interval) on changes in CRF in women with breast cancer (BC) receiving chemotherapy. A systematic review and meta-analysis were conducted as per PRISMA guidelines, and randomized controlled trials comparing usual care (UC) and aerobic training in women with BC undergoing chemotherapy were eligible. The results suggest that increases in CRF are favored by (a) aerobic training when compared to usual care; (b) vigorous-intensity aerobic exercise (64–90% of maximal oxygen uptake, VO_2max_) when compared to moderate-intensity aerobic exercise (46–63% of VO_2max_); and (c) both continuous and interval aerobic training are effective at increasing the VO_2max_. Aerobic training improves CRF in women with BC undergoing chemotherapy. Notably, training intensity significantly impacts the VO_2max_ response. Where appropriate, vigorous intensity aerobic training should be considered for women with BC receiving chemotherapy.

## 1. Introduction

According to the World Health Organization (WHO), cancer was the leading cause of death worldwide in 2018. Two million new cases and over 600,000 deaths were attributed to breast cancer (BC) alone [1]. In BC patients, anticancer therapy most commonly involves chemotherapy and is considered crucial in improving survival [2]. Despite its positive clinical effect, chemotherapy has been associated with debilitating side effects [3], such as muscle atrophy [4,5], cancer-related fatigue [6,7,8], and cardiotoxicity [9,10,11]. Further, a 31% impairment in cardiorespiratory fitness (CRF), as measured by the peak/maximal oxygen consumption (VO_2max_), has been observed in women with BC undergoing adjuvant chemotherapy [12]. This is a concern, given that emerging evidence suggests that CRF is a significant prognostic marker, with data indicating that a poor VO_2max_ is associated with a poorer quality of life, treatment-induced cardiotoxicity, and an increased risk of cancer-related mortality [13,14,15,16].

Aerobic training has been shown to improve CRF and other cancer-related health outcomes in cancer patients [17,18], which has led major health organizations to include aerobic exercise in their physical activity guidelines for cancer patients [19,20]. However, the efficacy of aerobic training on CRF specifically in women with BC undergoing chemotherapy has not yet been systematically assessed, and it remains unclear what the most effective training protocols are with regard to training intensity and mode. For instance, Segal et al. (2001) [21] compared a continuous aerobic exercise guideline-based training group (chemotherapy + 150 min of moderate (50–60% of VO_2max_) continuously walking) versus an usual care (UC) non-exercising group. The authors showed that, after 26 weeks of training, the VO_2max_ decreased by 0.3 mL∙kg^−1^min^−1^ in the training group and increased by 0.2 mL∙kg^−1^min^−1^ in the UC group. Conversely, Jones et al. [22] showed that 12 weeks of moderate/vigorous-intensity aerobic training (55–100%-VO_2max,_ 20–45 min-, 3 times/week) increased the VO_2max_ (from 19.5 ± 7.6 to 22.1 ± 7.0 mL∙kg^−1^min^−1^) in women with BC undergoing chemotherapy. Taken together, these results suggest that aerobic training intensity may play a role when aiming to improve VO_2max_ in women with BC undergoing chemotherapy. Considering aerobic training mode, both continuous and interval aerobic training have shown to successfully impact cardiac function and hence CRF in BC patients undergoing chemotherapy [22,23,24]. Thus, identifying the most effective aerobic training protocols to increase VO_2max_, a strong predictor of symptom burden; cardiac function; [12] and mortality, may help to mitigate short- and long-term health issues in BC survivors.

In order to account for differences in aerobic training protocols between studies and factors such as low statistical power (small sample sizes) in randomized controlled trials, we conducted a meta-analysis. A meta-analysis summarizes data from different studies and allows testing for the effects of moderator variables (e.g., exercise intensity, mode of aerobic training), thereby improving the estimated precision of the factors affecting the changes in VO_2max_. The purpose of this systematic review and meta-analysis was to determine: (a) the efficacy of aerobic training to increase CRF (measured by VO_2max_); (b) the role of aerobic training intensity (moderate or vigorous) on the VO_2max_ response; (c) the effect of the aerobic training mode (continuous or interval) on the changes in VO_2max_ in women with BC receiving chemotherapy.

## 2. Methods

### 2.1. Protocol and Registration

The systematic review was registered with PROSPERO (CRD42019134584), and the Preferred Reporting Items for Systematic Reviews and Meta-Analyses guidelines for systematic review reporting were followed [25].

### 2.2. Search Strategy and Information Sources

A systematic literature review was performed using three major databases (PubMed, Scopus, and Web of Science). The last search was performed on November 1st, 2019. This search was applied with no limits to publication year, type, and status. The MeSH terms were combined as follows: ((((“breast carcinoma” [Title/Abstract] OR “breast neoplasm” [Title/Abstract] OR “breast tumor” [Title/Abstract] OR “breast cancer” [Title/Abstract] OR “mammary cancer” [Title/Abstract] OR “mammary carcinoma” [Title/Abstract]))) AND ((chemotherapy [Title/Abstract] OR “adjuvant chemotherapy” [Title/Abstract] OR “neoadjuvant chemotherapy” [Title/Abstract] OR chemoradiotherapy [Title/Abstract] OR radiochemotherapy [Title/Abstract] OR “neoadjuvant therapy” [Title/Abstract] OR “adjuvant therapy” [Title/Abstract]))) AND ((“exercise training” [Title/Abstract] OR “aerobic exercise” [Title/Abstract] OR “training exercise” [Title/Abstract] OR exercise [Title/Abstract] OR “physical exercise” [Title/Abstract] OR “endurance exercise” [Title/Abstract] OR endurance [Title/Abstract] OR “high-intensity interval training*” [Title/Abstract] OR HIIT [Title/Abstract] OR “sprint interval training*” [Title/Abstract] OR “physical activity” [Title/Abstract])). Two researchers conducted the review independently (GM and HB). Discrepancies between researchers were discussed and, if necessary, a third researcher was consulted (MSC). Reference lists of identified articles were also searched for additional relevant articles on the topic.

### 2.3. Eligibility Criteria

Only English-language studies, Pilot Studies (PS), and Randomized Controlled Trials (RCT) comparing the effects of aerobic training (continuous or interval; home-based or under professional supervision) and usual care on the VO_2max_ of women with BC undergoing chemotherapy (adjuvant or neoadjuvant) were included. Studies that included participants of any age with histologically confirmed BC undergoing chemotherapy in combination with other treatments were considered (e.g., radiotherapy or hormonal therapy). Studies that evaluated different types of cancer (e.g., breast, ovarian, rectal, etc.) were only included if data from a breast cancer-only group were provided. The primary outcome for this systematic review and meta-analysis was CRF, measured by the maximum oxygen uptake (VO_2max_ or VO_2peak_). Training intensity and mode were used as moderator variables.

The exclusion criteria were as follows: (i) duplicated articles; (ii) duplicated data; (iii) articles without original data (e.g., comments, reviews, case reports, and technical reports); (iv) studies with dietary counseling or intervention; (v) studies involving metastatic breast cancer patients; and (vi) studies with no usual care-only group.

### 2.4. Study Selection and Data Extraction

To reduce the selection bias potential, the titles and abstracts of all studies were independently evaluated by two investigators (GM and HIB). Duplicated studies were excluded, and the remaining ones were screened, assessed for eligibility criteria, and then forwarded to data extraction. Data extraction was performed by three independent reviewers (GM, HIB, and MSC) for the following variables: authors, year of publication, sample size, treatment protocol, exercise protocol, and pre and post-intervention mean ± standard deviation (SD) values of the VO_2max_/VO_2peak_. The data extraction procedures were standardized using a pre-piloted Excel spreadsheet. To test for possible coding drift, we randomly selected 30% of the studies for recoding following procedures outlined by Cooper et al. [26]. The mean agreement between coders was 95%.

### 2.5. Assessment of Risk of Bias within and across Studies and Quality

The risk of bias of the included studies was independently assessed by two authors (CU and MSC) using the Revised Cochrane risk-of-bias tool for Randomized Trials (RoB2) [27]. The following five domains were assessed: {1} bias arising in the randomization process; {2} bias due to deviations from intended interventions; {3} bias due to missing outcome data; {4} bias in the measurement of the outcome and {5} bias in the selection of the reported result. Each domain was classified as having {1} LOW risk of bias; {2} SOME CONCERN of risk of bias and {3} HIGH risk of bias. The Revised Cochrane risk-of-bias data is presented in Figure 1.

### 2.6. Training Protocol Classification

Each study was reviewed regarding the prescribed aerobic training protocol. Some studies implemented a periodized aerobic training protocol (i.e., progressive increase in training load), while others used a constant relative training intensity throughout the experimental period. To determine the predominant training intensity, we identified the intensity in which patients performed most of the training sessions. Training intensities were classified as moderate or vigorous. We used the American College of Sports Medicine (ACSM) equivalence table to determine the relative training intensity [28] of all the studies (Table 1). Further, the training protocols were classified as either interval or continuous training, and the total minutes of training were determined. Warm-up and cool down minutes were not included when computing the total training minutes. 

It is important to highlight that seven out of nine studies measured the VO_2max_ directly. Mijwel et al. 2018 [29] estimated the VO_2max_ using a submaximal cycle ergometer test. Ma 2018 [23] indirectly measured VO_2max_ using a treadmill exhaustion protocol. Seven studies measured VO_2_ directly using an exhaustion protocol on a cycle ergometer (Jones et al. 2013 [22]; Lee et al. 2019 [24]; Moller et al. 2015 [30]; Mowafy et al. 2016 [31]) or treadmill (Al-Majid et al. 2015 [32]; Courneya et al. 2007 [33]; Kim et al. 2006 [34]). It is also necessary to report that five studies conducted the VO_2max_ measurement and the training on the same device [22,23,24,31,32], while four other studies measured the VO_2max_ and conducted the training on different devices [29,30,33,34].

## 3. Statistical Analysis

Initially, pre- to post-training effect sizes (*ES*) were calculated according to Equation (1). Then, the between-group difference effect sizes (*d*) for the dependent variable (i.e., maximum oxygen uptake (VO_2max_)) were calculated according to Equations (2–5) for each study. We estimated the pre- to post-correlation for VO_2max_ and its respective confidence interval using a bootstrapping approach based on previously published [35] and unpublished data from our group. In order to be conservative, we used the lower limit of the confidence interval obtained from the bootstrapping estimate of the pre- to post-training correlation for all studies (*r* = 0.611). The heterogeneity for between-study variability was verified with the *I*^2^ statistics, with thresholds set as *I*^2^ = 25% (low), *I*^2^ = 50% (moderate), and *I*^2^ = 75% (high) [36]. Due to the high between-study heterogeneity, the data were analyzed using a random-effect model.

To investigate the potential effect of exercise intensity on the VO_2max_, we converted the exercise intensity into a categorical variable with two levels—I) low to moderate intensity; and II) vigorous intensity—which was included as a moderator variable. Similarly, the training mode was included as a categorical variable with two levels: I) continuous and II) interval training. The total volume (i.e., minutes per week) and training intervention period (i.e., the total number of training weeks) were included as continuous moderator variables in the meta-regression models. A sensitivity analysis, removing one study at a time and re-analyzing the summary effect, was performed to identify possible highly influential studies. Studies were considered influential if the removal significantly changed the summary effect (i.e., change going from significant to non-significant) [37]. Publication bias was verified using the funnel plot, Kendall’s tau with continuity, and Egger’s regression approaches. In cases of significant publication bias, the fill and trim procedure was implemented. All the data were analyzed using the *rma* and forest functions available in the metafor package for Rstudio (Version 3.6.3). The significance level was set at *p* < 0.05. Data are presented as the mean ± standard error or standard deviation and confidence interval.
(1)$$ES=\frac{Mean_{post}-Mean_{pre}}{SD_{pre}},$$
(2)$$d=C\left[\frac{\left(Mean_{posttreatment}-Mean_{pretreatment}\right)-\left(Mean_{postcontrol}-Mean_{precontrol}\right)}{SD_{changepooled}}\right],$$
(3)$$SD_{changepooled}=\sqrt{\frac{\left(n_{{treatment}^{-1}}\right)SD_{changetreatment}^{2}+\left(n_{{control}^{-1}}\right)SD_{changecontrol}^{2}}{n_{treatment}+n_{{control}^{-2}}}},$$
(4)$$C=1-\frac{3}{4\left(n_{treatment}+n_{{control}^{-2}}\right)},$$
(5)$$SD_{change}=\sqrt{SD_{pre}^{2}+SD_{post}^{2}-2\cdot r_{pre-postcorrelation}\cdot SD_{pre}\cdot SD_{post}}.$$

## 4. Results

The initial search returned 1967 studies, and 998 duplicated studies were removed. For the remaining studies (969), the titles and abstracts were screened and 919 were excluded. The remaining 50 studies were assessed for eligibility, considering our inclusion criteria. Forty-one studies were excluded, and nine studies were included in the systematic review and meta-analysis. The search and study selection process is depicted in Figure 2.

The present meta-analysis included nine studies that compared structured aerobic training vs. usual care (i.e., no exercise), which resulted in nine treatment outcome measures. The overall number of participants in the nine studies were 493. The average pre- to post-intervention change in the VO_2max_ was 9.97% (ES: 0.62 _95%_CI: −0.29 to 1.53) and −10.18% (ES: −0.54 _95%_CI: −0.99 to −0.10) for the training and usual care groups, respectively. The overall between-group effect size difference for the VO_2max_ favored the aerobic training group (*d*: 1.19 ± 0.38 _95%_CI: 0.45 to 1.94) (Figure 3).

The subgroup analyses revealed a significant effect favoring vigorous-intensity aerobic training protocols in increasing VO_2max_ (Table 2). Regarding the training mode/type, both continuous and interval aerobic training significantly increased the VO_2max_.

The meta-regression analyses did not produce significant betas for both continuous variables: total session exercise volume and total intervention period (*p* > 0.05). The sensitivity analysis demonstrated that CRF was not highly affected by any of the individual studies. Visual inspection of the funnel plot showed seven studies outside the funnel limits (five in the left and two in the right); both Kendall’s tau with continuity correction (tau = 0.55; *p* = 0.04) and the Egger’s regression intercept did not show significant bias (*z* = 2.0019; *p* = 0.04) (Figure 4). As the fill and trim procedure did not change the effect size estimate, we maintained the initial analysis.

## 5. Discussion

Emerging evidence on the prognostic significance of CRF, measured by VO_2max_, underpins the clinical importance of developing effective strategies to prevent and/or recover low VO_2max_ in women with BC. Given that aerobic training is the upmost recommended exercise intervention to improve CRF, we conducted the first meta-analysis to assess: (a) the efficacy of aerobic training to increase VO_2max_; (b) the effect of moderate and vigorous intensity aerobic training on the VO_2max_ response; (c) the effect of the aerobic training mode (continuous or interval) on the changes in VO_2max_ in women with BC receiving chemotherapy. Our main results show that: (a) aerobic exercise training significantly increases the VO_2max_ compared with UC; (b) only vigorous-intensity aerobic exercise (64–90% of VO_2max_) significantly increases VO_2max_, with no effect for moderate-intensity aerobic protocols (46–63% of VO_2max_); and (c) both continuous and interval aerobic training are effective at increasing VO_2max_. Taken together, our results suggest that, where appropriate in clinical practice, vigorous aerobic training performed with continuous or interval training mode should be considered to improve the CRF in women with BC undergoing chemotherapy.

Previous studies [17,18] have demonstrated the feasibility, safety, and overall effectiveness of aerobic training for patients with BC undergoing chemotherapy; however, there is no consensus regarding the most appropriate training intensity and mode to optimize training-induced adaptations. Most studies involving BC patients have prescribed aerobic training based on guidelines from the Clinical Society of Oncology of Australia (COSA) [38], the American College of Sport Medicine (ACSM) [39], and the American Cancer Society (ACS) [40]. These guidelines recommend that patients with BC should perform 150 min of moderate or 75 min of vigorous intensity aerobic exercise per week. Our data support this recommendation, and three out of seven studies [23,31,32] (Figure 3) demonstrated that vigorous intensity aerobic training was effective in significantly improving the VO_2max_ compared to usual care. These findings are supported by a study that directly compared different exercise intensities in patients undergoing chemotherapy and found that moderate to vigorous intensity exercise led to more significant improvements in cardiorespiratory fitness and physical function and decreased the severity of adverse effects such as nausea, vomiting, pain, and physical fatigue to a greater degree than low-intensity exercise [41]. Altogether, these results suggest that, where appropriate, the prescription of vigorous intensity aerobic training should be considered for women with BC; however, additional randomized clinical trials are still necessary to substantiate both the safety and efficacy of these training protocols in larger cohorts.

Eight out of nine studies included in our review treated women with BC using anthracycline (AC)-based chemotherapy, which commonly induces cardiotoxicity, one of the most debilitating chemotherapy-related side effects [42,43,44]. To date, AC-induced cardiotoxicity most commonly presents as a decrease in the left ventricular ejection fraction (LVEF) [45] and, ultimately, heart failure [9,43]. Cardiotoxicity can occur at any time during AC infusion and up to years or decades later, known as late onset chronic cardiotoxicity [46,47]. A meta-analysis by Haykowsky et al. [48] showed that vigorous-to-maximal aerobic exercise was more effective than moderate-intensity exercise at improving the LVEF and VO_2max_ in patients with heart failure. A corollary from Haykowsky’s study is that women with BC should be encouraged to perform vigorous aerobic training to prevent/treat chemotherapy-associated decreases in LVEF. Accordingly, it has been shown that vigorous exercise can prevent toxicity-related reductions in chemotherapy dose, which is critical in order to restrain tumor growth [17,41]. Importantly, exercise protocols with vigorous intensities have shown to produce robust increases in VO_2max_ when compared to moderate intensity protocols, regardless of the equalization in training volume [49]. Additionally, the exercise intensity seems not to affect training adherence [50]. Accordingly, higher levels of physical fitness are associated with greater adherence [51,52] and inversely associated with fatigue levels in women with BC [53].

Vigorous or high-intensity aerobic training is normally performed as interval training [54]. It is well known that the longer an individual can exercise at intensities close to the minimum velocity of VO_2max_, the greater the gains in VO_2max_ appear to be [54,55,56]. Due to the nature of interval training, which includes short sets of vigorous to maximal exercise (≥90% of VO_2max_) interspaced by low-intensity recovery periods, the maintenance of training intensity during aerobic exercise is possible. Our moderator analysis suggests that both continuous and interval aerobic training are effective at increasing VO_2max_. There is some compelling evidence that vigorous intensity continuous aerobic training performed during chemotherapy counteracts cancer-related fatigue, which can last up to 12 months after treatment completion, and reduces the time to return to work as compared with the UC group [57]. However, we acknowledge that performing continuous aerobic training at vigorous intensities is very demanding and may not be feasible for women with BC undergoing chemotherapy, thus interval training should be considered as a viable alternative. Additionally, some studies have demonstrated that interval training is more effective than continuous training to at increasing VO_2max_ in heart failure patients with reduced ejection fraction [58] or coronary artery disease [59], which may be an important consideration for patients with chemotherapy-associated cardiotoxicity.

The present meta-analysis has some limitations. According to our classification, only two studies assessed moderate-intensity aerobic training, and only two studies investigated an interval training mode, which makes it hard to draw definitive conclusions. Moreover, the high risk of bias of the included trials needs to be considered (Figure 1), which seems to be mainly related to the difficulty of blinding the interventions in eight out of nine included trials. It is therefore important to point out that the bias assessment for those items does not reflect a low quality of study design, but expresses the inevitable bias introduced by the lack of blinding. Another source of bias in the analysis is the CRF assessment method. Two studies assessed VO_2max_ indirectly, thus there is a small chance of bias with regard to the prescribed training intensity and training-induced increase in VO_2max_. However, as the magnitude of the effect size in Ma 2018 [23] was large and small in Mijwel et al. 2018 [29], one may suggest that the likely bias is negligible or small. It is reasonable to suggest that the differences in effect sizes were due to the higher training volume and more intensive training regimen. Thus, larger and better blinding-controlled trials should be conducted to properly resolve this issue.

The data of the present manuscript indicate that high-intensity aerobic training can be performed by women with BC under chemotherapy regardless of the training method (continuous or interval training), however there are several considerations that need to be taken into account in clinical practice. High-intensity continuous training (walking/run, 5x week at 70% of VO_2peak_ for 30 min) may be very demanding for this cohort. Thus, performing high-intensity interval training (e.g., cycling 3x of 3min at 90% of VO_2peak_, with intervals cycling at 30% of VO_2peak_, 3x week) seems to be a more feasible alternative, as it is less time-consuming and more enjoyable. Further, while several studies prescribe exercise intensity based on heart rate, chemotherapy may influence a patient’s heart rate. Although heart rate can be easily monitored on a daily basis, and heart hate reserve can be calculated, the training intensity can be influenced. Hence, a watt-based training prescription represents a less error-prone method. On the other hand, there is mounting evidence that breast cancer chemotherapy protocols elicit high levels of fatigue, which can greatly decrease a patient’s ability to maintain a target Wattage. Finally, when the training sessions are not performed with the same equipment (e.g., stationary bike, treadmill) that is used for the CRF assessment, the training intensity may have been a little under due to non-specific training-induced peripheral adaptations (i.e., at the skeletal muscle level).

## 6. Conclusions

In summary, our findings indicated that aerobic exercise increases the VO_2max_ in women with BC undergoing chemotherapy. We also showed that vigorous intensities (64–90% of VO_2max_) performed with continuous or interval aerobic training are effective at increasing the VO_2max_ in women with BC undergoing chemotherapy. Performing continuous aerobic training at vigorous intensities is very demanding and, thus, interval aerobic training should be considered as a viable option for women with BC. While this work supports the benefits of aerobic exercise, additional clinical investigations are warranted to determine the effects of different exercise modalities, timings, and durations and to identify optimal aerobic training regimens to not only improve CRF but also counteract treatment-related side-effects, such as cardiotoxicity, in women with BC.

## Figures and Tables

**Figure 1 cancers-12-02240-f001:**
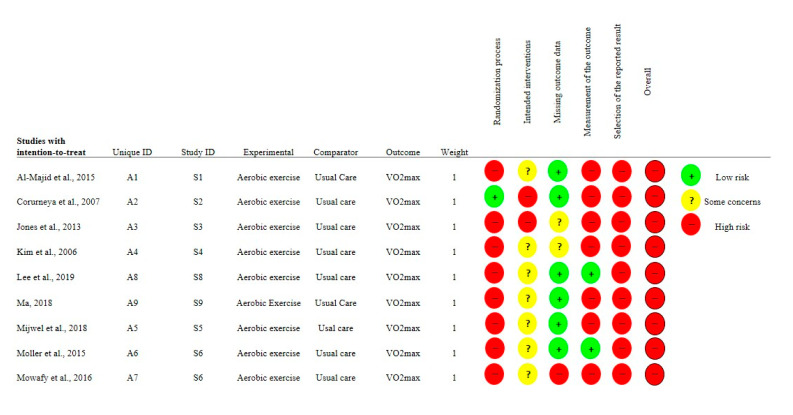
Risk of bias assessment for the studies included in the meta-analysis. Footnotes: AE = Aerobic Training; UC = Usual Care.

**Figure 2 cancers-12-02240-f002:**
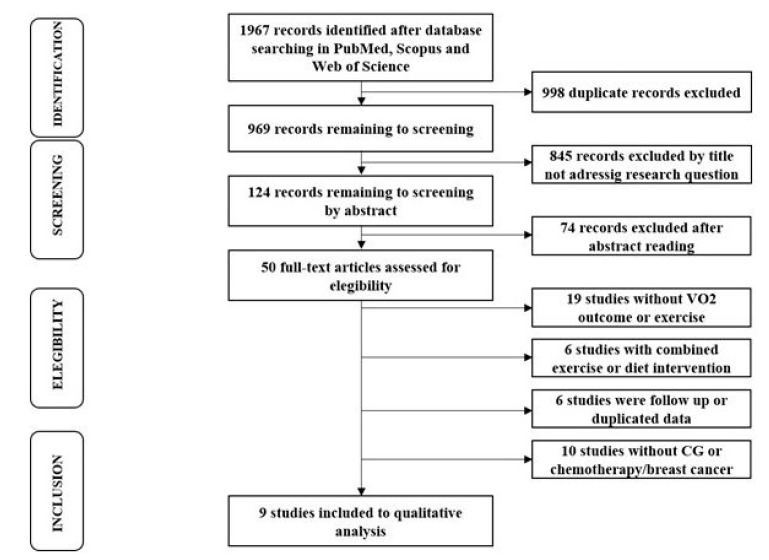
Flow chart of the search process.

**Figure 3 cancers-12-02240-f003:**
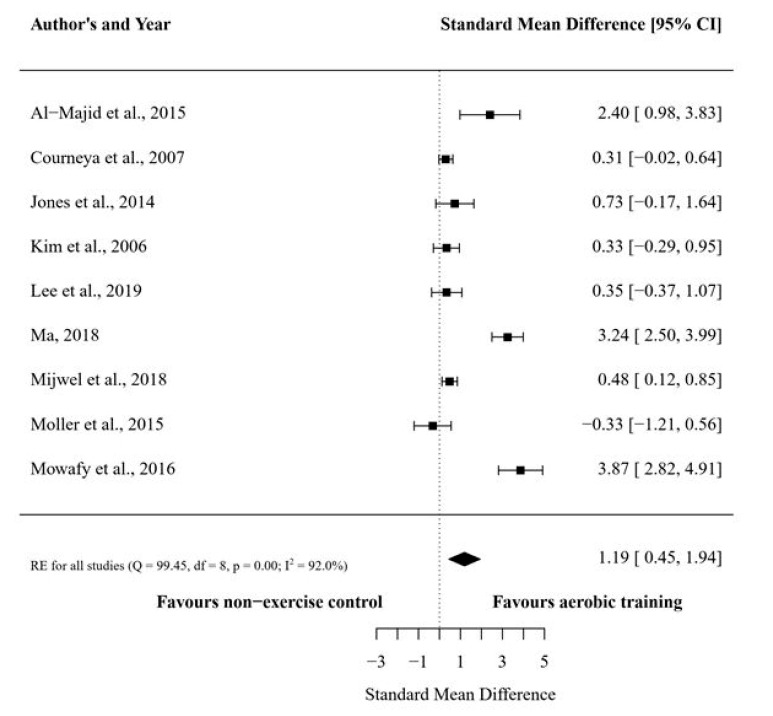
Forest plot for maximum oxygen consumption (VO2max) between the structured aerobic training groups and usual care (no exercise) groups. Footnotes: data are shown as between-group effect size difference (d) and 95% confidence interval.

**Figure 4 cancers-12-02240-f004:**
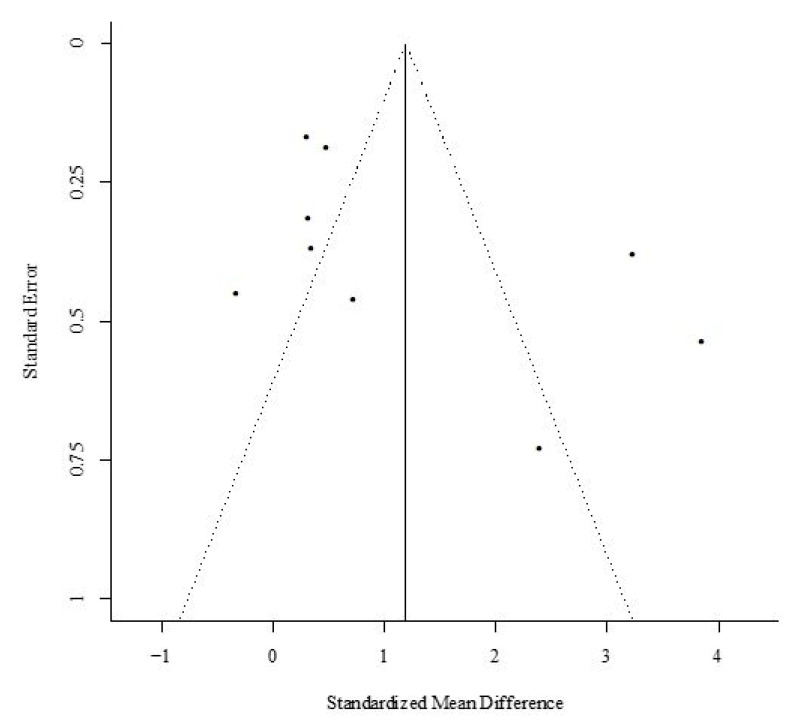
Funnel plot of studies comparing the maximum oxygen consumption (VO2max) between the structure aerobic training and usual care (no exercise) groups.

**Table 1 cancers-12-02240-t001:** Descriptive data of the studies included in the meta-analysis.

	Al-Majid et al. [32]	Courneya et al. [33]	Jones et al. [22]	Kim et al. [34]	Lee et al. [24]	Ma [23]	Mijwel et al. [29]	Moller et al. [30]	Mowafy et al. [31]
PUBLICATION YEAR	2015	2007	2013	2006	2019	2018	2018	2015	2016
AGE									
USUAL CARE	52.7 ± 10.7	49	46 ± 11	48.3 ± 8.8	44.7 ± 11.2	43.5 ± 6.3	52.6 ± 10.2	46.95 ± 9.19	45
TRAINING	47.9 ± 10.4	49	51 ± 6	51.3 ± 6.7	49.1 ± 7.9	44.2 ± 5.7	54.4 ± 10.3	48.49 ± 8.41	45
(N)									
USUAL CARE	7	73	10	19	15	33	51	10	20
TRAINING	6	71	10	22	15	31	70	10	20
CHEMOTHERAPY									
NEO			Y		Y				
ADJ	Y	Y		Y	Y	Y	Y	Y	Y
TYPE									
NON TAX									
AC		Y	Y	Y	Y	Y	Y	Y	
CYC			Y					Y	
FE100C, CE120F		Y							
TRAST									
TAXANE		Y		Y			Y	Y	
TRAINING PROTOCOL									
DURATION (weeks)	12	18	12	8	8	16	16	12	16
MODE	CON	CON	CON	CON	INT	INT	CON	CON	CON
VOLUME MEAN (min/week)	90	97.5	102	90	63	150	60	150	45
INTENSITY	VIG	VIG	MOD	VIG	VIG	VIG	VIG	MOD	VIG
TIME (min) × INTENSITY									
LIGHT					336				-
MODERATE	130	315	795			574		1800	-
VIGOROUS	1050	1170	400	720		960	802		-
MAXIMAL			30		168				
VO2 OUTCOMES									
USUAL CARE									
PRE	23.8 ± 2.9 *	24.8 ± 6.2 *	17.5 ± 4.8 *	1597 ± 357 †	18.7 ± 7.1 *	1210 ± 258 †	2.19 ± 0.53 ‡	30.5 ± 5.0 *	21.1 ± 2.5 *
POST	17.5 ± 2.8	23.5 ± 5.4	16.0 ± 4.0	1630 ± 351	16.1 ± 6.0	984 ± 157	1.94 ± 0.52	27.7 ± 6.8	21.0 ± 2.4
TRAINING									
PRE	26.1 ± 2.6	25.2 ± 7.2	19.5 ± 7.6	1671 ± 349	19.7 ± 8.7	1134 ± 268	2.10 ± 0.47	27.1 ± 6.4	21 ± 2.5
POST	26.0 ± 2.5	25.7 ± 7.4	22.1 ± 7.0	1810 ± 369	19.4 ± 6.6	1594 ± 190	2.06 ± 0.45	22.4 ± 6.5	30.8 ± 3.5

NEO = neoadjuvant; ADJ = adjuvant; TAX = taxane; AC = anthracycline; CYC = cyclophosphamide; TRAST = trastuzumab; CON = continuous; INT = interval; VIG = vigorous; MOD = moderate. LIGHT = 37–45% of VO_2max_; MODERATE = 46–63% of VO_2max_; VIGOROUS = 64–90% of VO_2max_; MAXIMAL = ≥91% of VO_2max_. * Unit: mL·kg^−1^·min^−1^; † mL·min^−1^; ‡ L·min^−1^. It was not possible to identify the time that was performed at each training intensity; however, as the training was performed until exhaustion, it was classified as vigorous.

**Table 2 cancers-12-02240-t002:** Effects of training intensity and training mode on the maximum oxygen consumption (VO_2max_) compared to usual care (no exercise) subgroup analyses.

Subgroup	*N*° participants	*d* (_95%_CI)	*p* Value
Training intensity			
Low- to moderate [22,30]	20	0.20 (−1.44 to 1.85)	0.81
Vigorous [23,24,29,31,32,33,34]	235	1.47 (0.60 to 2.34)	0.0009
Training mode			
Continuous [22,29,30,31,32,33,34]	209	1.01 (0.19 to 1.83)	0.0157
Interval [23,24]	46	1.79 (0.28 to 3.29)	0.02

lower to moderate: 40–59% of heart rate reserve; vigorous: 60–89% of heart rate reserve; *d*: between-group effect size difference; *I*^2^: heterogeneity for between-studies variability.

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
