# Peer review of "Aerobic Exercise-Induced Changes in Cardiorespiratory Fitness in Breast Cancer Patients Receiving Chemotherapy: A Systematic Review and Meta-Analysis"

_cancers, 2020, doi:10.3390/cancers12082240_

Round 1

Reviewer 1 Report

The authors carried out a systematic review and metanalysis on aerobic exercise-induced changes in cardiorespiratory fitness in breast cancer patients receiving chemotherapy.

Although literature provides a wide range of papers on this topic, the authors reported on cardiorespiratory fitness that is not always considered as an objective outcome measure. Methoddology is appropriate and results are interesting, I would suggest to stress more their eventual translation into clinical practice.

I think the manuscript is worth being published.

Reviewer 2 Report

Thank you for the opportunity to review this manuscript. It is a very well written manuscript. It addresses an important health issues of women with breast cancer. The authors put great efforts on describing the detailed steps of review and the meta-analysis; thus, sounds reliable and easy to follow the context.

I have two suggestions for the authors:

  • Table 1:
    Please add units of the values. For example, “min” for “usual care” values.
  • Under "Introduction" or "Discussion" sections:
    Please provide some of the examples of continuous and interval aerobic training.

Reviewer 3 Report

Dear authors,

you conducted a systematic review of exercise trials in order to determine

  1. the efficacy of aerobic exercise to improve CRF - (reflected by VO2 max)
  2. the role of training intensity (moderate to vigorous)
  3. the effect of training mode  (continuous - interval)

After intensive literature research 9 studies meet their selection criterias of randomized trials with training and usual care groups with a total number of 493 patients.

From the scientific point of view it would be interesting to answer the question if structured training during antineoplastic treatment is feasible and able to maintain muscle mass and cardiorespiratory fitness, however, as you have been able to demonstrate: many limitations complicate this scientific issue.

Some sudies included a very limited number of patients and bias (figure 1)

- randomisation bias  - in table 1 significant differences of Vo2max at study entry between training and usual care - reasons? what is the potential influence on your scientific question.

Since VO2 max plays a key role for answering the scientific question:Give information on the testing system of VO2max (measured or calculated - bicycle or treadmill spiroergometry) and if the form of aerobic exercise was selected by the testing system (background: maximum heart rate depends on the training form - training intensity was based on heart rate reserve)

Is there any information on maximum loading available? If any of these patients fulfill maximum load criteria?

Heart rate is very variable and it might be or probably is influenced by cortisone, other concomitant medication or chemotherapy itself. What do the authors think, why training was monitored by heart rate? (Alternative: watt based)

It is not surprising that aerobic training below 60% of heart rate reserve do not result in maintaining/gaining cardiorespiratory fitness: it is well known that aerobic training for improvement of cardiorespiratory fitness has to be at least 60% of heart rate reserve - no additional or surprising result

Please redesign table 1: confusing

- changes in reference values VO2 max - * Training protocol * HRR 30-39%

Table 2 subgroup analysis: pls give information on number of patients instead of number of trials

Round 2

Reviewer 3 Report

Dear authors,

Thank you for your detailed response to my critical comments and that you have included some changes. To be congratulated to your efforts, the article has been improved to some extent. 

As you have been able to underline with your meta-analysis exercise is able to maintain/increase maximum oxygen uptake in patients receiving chemotherapy. 

Some issues still remain:

Outcome parameter VO2 max:

In two studies calculated, method not mentioned -limitation of calculation - signs of maximum load 

Different reference values (?ml/min/kg; l/min, ml/min? Pls add reference values - Table 1)

Training:

monitoring of training 

Monitoring of training: based on % heart rate, % heart rate reserve, % watt power, %VO2 max: many options are available and used in the original studies/trials - the authors have chosen hear rate reserve as common factor. Is there information on heart rate reserve of each study available?

Intervall training: 

Lee et al, 2019 (high intensity interval training, Lee at al 2019) chose peak power output - 7 times 1 minute 90 % of PPO! - In the original publication, there is no information on heart rate (reserve) provided.

Ma, 2018 monitored high intensity training by % 90-95 of maximum heart rate, 4x 5min, intermittent period 3 min with 50 - 70 % maximum heart rate

In view of the available results it is not possible to favor interval training and the suggested protocol  (line 297) hasn’t been proved in both included interval training studies/trials. Please describe advantages and disadvantages of each trainings method (interval - duration of interval; continuous)

Author Response

Dear authors,

Thank you for your detailed response to my critical comments and that you have included some changes. To be congratulated to your efforts, the article has been improved to some extent. 

As you have been able to underline with your meta-analysis exercise is able to maintain/increase maximum oxygen uptake in patients receiving chemotherapy. 

Some issues still remain:

In two studies calculated, method not mentioned -limitation of calculation - signs of maximum load 

Answer: Indeed, there is a chance of bias when determining the training effect as VO2max was assessed indirectly. However, we would like to highlight that the effect sizes (Figure 3) of the studies that assessed VO2 indirectly have similar magnitudes than the ones that assessed it directly. Thus, it is reasonable to suggest that a possible bias had a negligible/small effect in the overall results. We added some sentences in the discussion to address this issue (Lines 282 – 286).

Different reference values (?ml/min/kg; l/min, ml/min? Pls add reference values - Table 1)

Answer: The VO2max values presented on Table 1 are different because they are expressed using different units of VO2max (i.e. ml/kg/min; mL/min; L•min-1). Such a scenario is very usual in meta-analyses and is the main reason why results are presented as effect sizes (dimensionless). We added the reference values as suggested.

Training:

monitoring of training 

Monitoring of training: based on % heart rate, % heart rate reserve, % watt power, %VO2 max: many options are available and used in the original studies/trials - the authors have chosen hear rate reserve as common factor. Is there information on heart rate reserve of each study available?

Lee et al, 2019 (high intensity interval training, Lee at al 2019) chose peak power output - 7 times 1 minute 90 % of PPO! - In the original publication, there is no information on heart rate (reserve) provided.

Ma, 2018 monitored high intensity training by % 90-95 of maximum heart rate, 4x 5min, intermittent period 3 min with 50 - 70 % maximum heart rate

Answer: The reviewer has a good point. HRR values was not presented in all studies. For instance, Lee et al. 2019 measured VO2max in a maximal cycling exercise test and determined peak power output (PPO) during the last stage of an incremental test. Then, % PPO was used to prescribe training intensity. In the first version of our manuscript, we classified training intensity considering that PPO is linearly correlated with VO2max as showed by Muniz-Pumares et al. 2017[2], using the VO2max equivalence according to ACSM [1]. As not all of the included studies had HRR data, and the procedure that we used to obtain it may be error-prone, we changed the way that we did the training intensity classification. Thus, in this version of the manuscript, training intensity was classified as “vigorous” or “moderate” according to the VO2max data suggested by the ACSM (Please, see Lines 140-145). It is important to emphasize that the alteration of the description does not affect the classification of intensity training that each study received previously.

In view of the available results it is not possible to favor interval training and the suggested protocol  (line 297) hasn’t been proved in both included interval training studies/trials. Please describe advantages and disadvantages of each trainings method (interval - duration of interval; continuous)

Answer: We do agree with the reviewer. Our findings do not support greater effectiveness of interval training compared to continuous training. Accordingly, we concluded that both training modes, continuous or interval, have similar benefits (Line 306) to women with breast cancer undergoing chemotherapy. However, in our experience, performing continuous vigorous-intensity aerobic exercise would be very demanding to women with breast cancer undergoing chemotherapy. On the other hand, there is evidence in the literature that interval training protocols are more time-efficient and enjoyable than continuous training, in spite of increased difficulty to control the training intensity and work-to-rest ration. These ideas were added to the manuscript (Please, see Lines 291-295).

REFERENCES

  1. Garber, C.E.; Blissmer, B.; Deschenes, M.R.; Franklin, B.A.; Lamonte, M.J.; Lee, I.-M.; Nieman, D.C.; Swain, D.P. American College of Sports Medicine position stand. Quantity and quality of exercise for developing and maintaining cardiorespiratory, musculoskeletal, and neuromotor fitness in apparently healthy adults: guidance for prescribing exercise. Med. Sci. Sports Exerc. 2011, 43, 1334–1359, doi:10.1249/MSS.0b013e318213fefb.
  2. Muniz-Pumares, D.; Pedlar, C.; Godfrey, R.; Glaister, M. The effect of the oxygen uptake-power output relationship on the prediction of supramaximal oxygen demands. J. Sports Med. Phys. Fitness 2017, 57, 1–7, doi:10.23736/S0022-4707.16.05948-X.
  3. Hood, M.S.; Little, J.P.; Tarnopolsky, M.A.; Myslik, F.; Gibala, M.J. Low-volume interval training improves muscle oxidative capacity in sedentary adults. Med. Sci. Sports Exerc. 2011, 43, 1849–1856, doi:10.1249/MSS.0b013e3182199834.
  4. Buchheit, M.; Laursen, P.B. High-intensity interval training, solutions to the programming puzzle: Part I: cardiopulmonary emphasis. Sports Med. 2013, 43, 313–338, doi:10.1007/s40279-013-0029-x.
